# Foldy: An open-source web application for interactive protein structure analysis

**Jacob B. Roberts** [1,2,3‡], **Alberto A. Nava** [1,2,4‡], **Allison N. Pearson** [1,2,5], **Matthew R. Incha** [1,2,5], **Luis E. Valencia** [1,2,4], **Melody Ma** [6], **Abhay Rao** [3], **Jay D. Keasling** [1,2,3,4,7,8] *

1 Joint BioEnergy Institute, Lawrence Berkeley National Laboratory, Emeryville, California, United States of America, 2 Biological Systems and Engineering, Lawrence Berkeley National Laboratory, Berkeley, California, United States of America, 3 Department of Bioengineering, University of California, Berkeley, Berkeley, California, United States of America, 4 Department of Chemical and Biomolecular Engineering, University of California, Berkeley, Berkeley, California, United States of America, 5 Department of Plant and Microbial Biology, University of California, Berkeley, Berkeley, California, United States of America, 6 Department of Molecular and Cell Biology, University of California, Berkeley, Berkeley, California, United States of America, 7 Center for Synthetic Biochemistry, Shenzhen Institutes for Advanced Technologies, Shenzhen, People's Republic of China, 8 The Novo Nordisk Foundation Center for Biosustainability, Technical University Denmark, Kemitorvet, Denmark

‡ These authors share first authorship on this work.
* jdkeasling@lbl.gov

**Data Availability Statement:** The website is built with React and Flask, and an assortment of bioinformatics tools, all of which are available in Docker files which are freely available at https:// github.com/JBEI/foldy. The LBNL Foldy instance

## Abstract

Foldy is a cloud-based application that allows non-computational biologists to easily utilize advanced AI-based structural biology tools, including AlphaFold and DiffDock. With many deployment options, it can be employed by individuals, labs, universities, and companies in the cloud without requiring hardware resources, but it can also be configured to utilize locally available computers. Foldy enables scientists to predict the structure of proteins and complexes up to 6000 amino acids with AlphaFold, visualize Pfam annotations, and dock ligands with AutoDock Vina and DiffDock.

In our manuscript, we detail Foldy's interface design, deployment strategies, and optimization for various user scenarios. We demonstrate its application through case studies including rational enzyme design and analyzing proteins with domains of unknown function. Furthermore, we compare Foldy's interface and management capabilities with other open and closed source tools in the field, illustrating its practicality in managing complex data and computation tasks. Our manuscript underlines the benefits of Foldy as a day-to-day tool for life science researchers, and shows how Foldy can make modern tools more accessible and efficient.

## Author summary

Foldy is a cloud-based application that enables scientists to use AI-based structural biology tools such as AlphaFold and DiffDock without software expertise. With many different deployment options, it can be set up by individuals, labs, universities, and companies in the cloud with no need for hardware resources. Foldy can predict the structure of proteins and complexes up to 6000 amino acids, visualize Pfam annotations, and dock ligands

(https://foldy.lbl.gov) has been configured to make some structures visible to the public. For security purposes, "viewers" must be authenticated by logging in with any Google / GMail email account.

**Funding:** This work was part of the DOE Joint BioEnergy Institute (jbei.org) supported by the U.S. Department of Energy, Office of Science, Office of Biological and Environmental Research, through contract DE-AC02-05CH11231 between Lawrence Berkeley National Laboratory and the U.S. Department of Energy and by a Department of Energy Office of Science Distinguished Scientist Award to J.D.K. J.B.R. was supported in part by a fellowship award under contract [FA9550-21-F-0003] through the National Defense Science and Engineering Graduate (NDSEG) Fellowship Program, sponsored by the Air Force Research Laboratory (AFRL), the Office of Naval Research (ONR) and the Army Research Office (ARO). A.A.N. was supported by a National Science Foundation Graduate Research Fellowship, fellow ID [2018253421]. The views and opinions of the authors expressed herein do not necessarily state or reflect those of the United States Government or any agency thereof. Neither the United States Government nor any agency thereof, nor any of their employees, makes any warranty, expressed or implied, or assumes any legal liability or responsibility for the accuracy, completeness, or usefulness of any information, apparatus, product, or process disclosed, or represents that its use would not infringe privately owned rights. The funders had no role in study design, data collection and analysis, decision to publish, or preparation of the manuscript.

**Competing interests:** I have read the journal's policy and the authors of this manuscript have the following competing interests: J.D.K. has relationships with and financial interests in Lygos, Demetrix, Apertor Labs, Ansa Biosciences, Cyklos Materials, Elsevier, Center for Biosustainability at Technical University of Denmark, Shenzhen Institute of Advanced Technology, Napigen, Kalion, Praj, Zero Acre Farms, ResVita Bio, Orbillion, and Keasling Consulting. J.B.R. has financial interests in AlkaLi Labs.

with AutoDock Vina and DiffDock. Some structures are visible to the public on the Lawrence Berkeley Labs Foldy instance, and can be viewed at https://foldy.lbl.gov.

Our manuscript highlights the user interface, deployment options, relative strengths of Foldy compared to existing tools, and some past applications of Foldy. It's an accessible solution for researchers who are not software experts. Many deployment options are possible and we highlight two: one of which can be set up in minutes, and the other can handle the traffic of thousands of users and hundreds of thousands of protein structures and docked ligands. This makes advanced AI-based tools more widely available, paving the way for accelerating life science research.

By developing an easy-to-use platform, our work demonstrates that even computationally expensive AI-based tools like AlphaFold can be made accessible to a wide audience. Improvements in the accessibility of computational tools will allow more biologists to more easily apply computational tools to more problems. We are hopeful that Foldy addresses the growing need of making revolutionary computational tools accessible to more researchers.

## Introduction

Recent advances in machine learning have led to the development of highly accurate protein structure prediction methods [1–4]. These methods have produced impressive results in numerous applications including *de novo* protein design [5] and protein-protein interaction screening [6]. However, the steep requirements for storage space, GPU processing power, and RAM make the direct application of these tools difficult for many end users. There have been a number of both corporate and open source projects to make AlphaFold and other structural biology tools more accessible.

The projects with the greatest success in increasing AlphaFold's availability are ColabFold [7] and AlphaFold-Colab [8]. Both projects provide custom Google Colaboratory Jupyter notebooks, which utilize free compute resources hosted by Google Cloud or can be run on personally requisitioned virtual machines. These Jupyter notebooks provide an interactive mode of using AlphaFold without the need for any complex installation or configuration. However, there are several limitations to these notebooks including session timeouts, limited GPU power, and limited batch processing capabilities. They also do not provide a solution for storing and sharing structural predictions, or for running other processes like domain annotations or small molecule docking.

A few corporations have created webtools which are smoother than ColabFold and offer the management of your structures, as well as small molecule docking, but these tools are closed source and, by virtue of being privately owned, are subject to changing pricing models. Nvidia created the BioNeMo service to make AI more accessible to life science researchers, but it is a private implementation and currently only available to a few biotechnology companies. Benchling incorporates AlphaFold predictions but only for small structures [9]. LatchBio [10] and NeuroSnap [11] both offer AlphaFold and other structural biology tools as a paid service with generous initial pricing models, but their source code is private and their availability is subject to change.

Here we present Foldy, an easy-to-deploy and easy-to-use modern web app for folding a protein (AlphaFold [1]), predicting domain annotations (Pfam [12]), and docking small molecule ligands (AutoDock Vina [13] or DiffDock [14]). Its primary objectives are to provide an intuitive interface, facilitate deployment, and enable prediction tasks for tens to thousands of

users per instance. The integrated tools within Foldy facilitate a seamless transition between protein structure prediction and downstream analysis. It can be rapidly deployed in a cloud environment, and also offers the possibility of a hybrid deployment utilizing local computational resources. The design of the Foldy architecture aims to lower barriers to entry for both end users and institutions.

## Methods

We introduce Foldy's interface, deployment options, and scalability considerations.

### Interface

Foldy has four main views: the New Structure view, the Dashboard, the Tag view, and the Structure view. The New Structure view is where users can submit new structure prediction tasks. At a minimum, users must provide an amino acid sequence and a name for the structure. The Dashboard serves as the app's landing page, providing access to all other pages. By default, the Dashboard displays a table of the user's structures, but a search bar allows users to filter structures by name, user, protein sequence, or tag. The Tag view displays all structures with a particular tag, and exposes bulk tasks such as downloading structures or docking small molecule ligands.

Foldy supports user authentication with OAuth. For Foldy deployments with user authentication enabled, there are three user roles: viewer, editor, and admin. Viewers are any user with a Google account, and are allowed view-only access to structures which have been explicitly marked "public" and their associated data (logs, docking runs). Editors have full read and write access. Admins can access special administrative views not discussed, including a raw database view and the RedisQueue admin panel. Users are authenticated by their Gmail account, and user types are flag controlled.

The Structure view has two columns: the predicted structure is on the left and a tool panel is on the right (Fig 1). For example, the structure panel might display a structure like the one

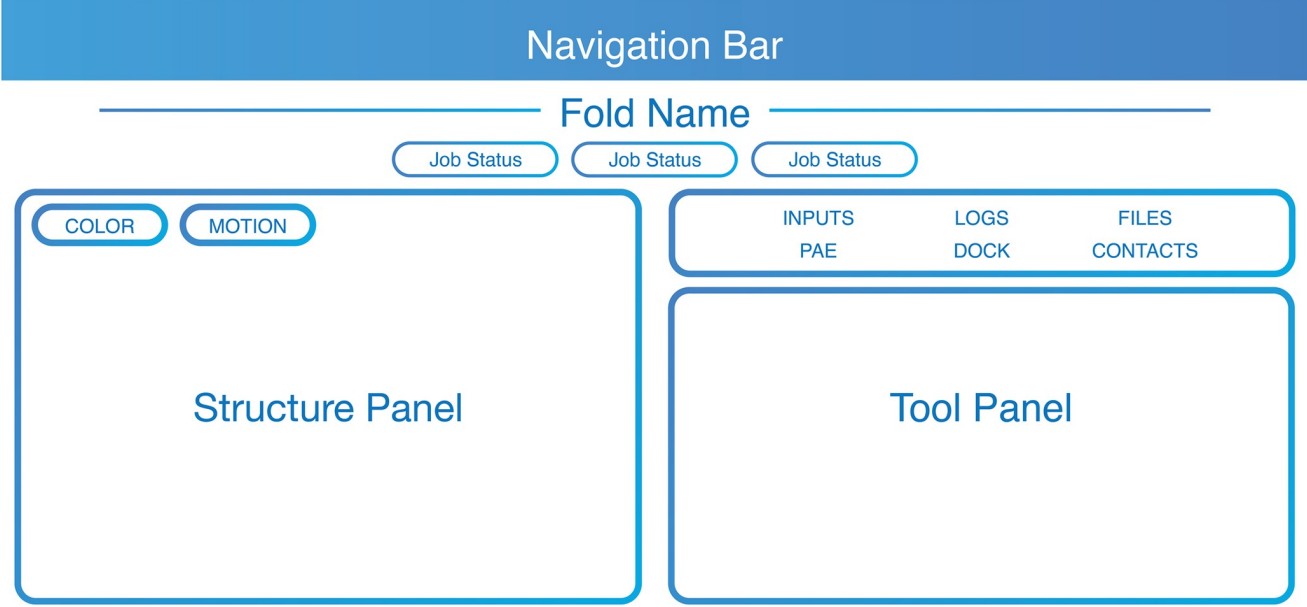

**Fig 1. Structure View.** The Structure view is a window into a predicted structure and a host of tools and associated information.

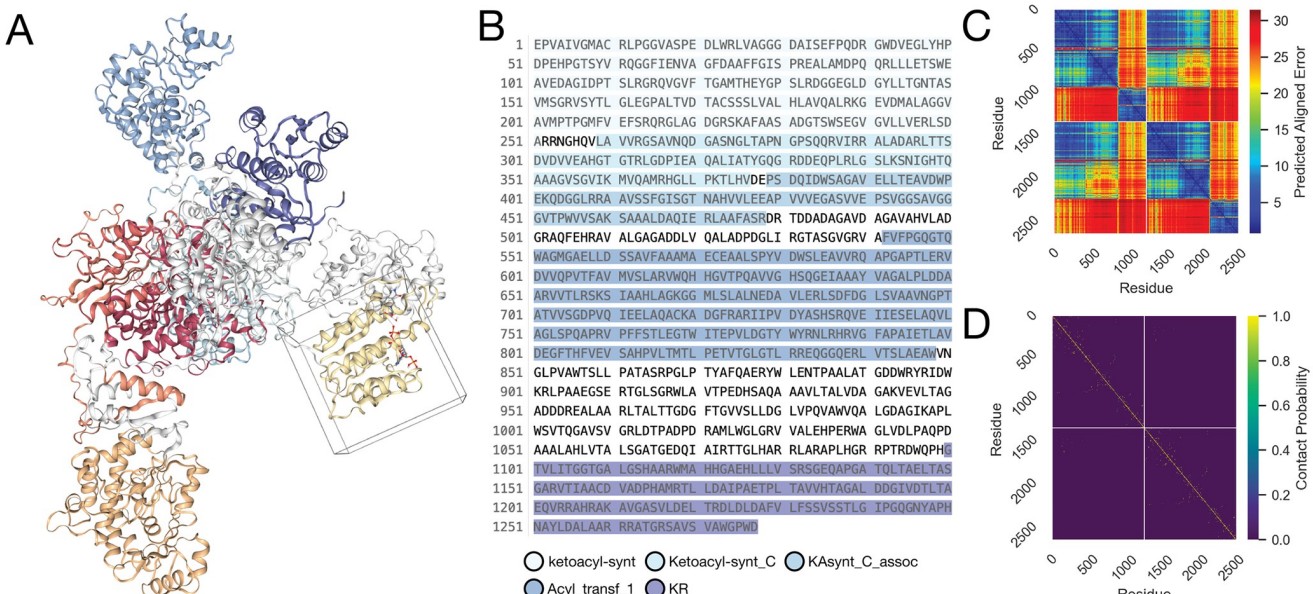

**Fig 2. Structure View Components.** *A. Structure Panel*: the left panel in the Structure View displays the structure. This is the predicted structure of a homodimer of the polyketide synthase pik2, with NADPH docked into the AT domain with AutoDock Vina. This structure was rendered with NGLViewer [15]. *B. Sequence View*: the amino acid sequence is displayed in the Inputs tab, and is optionally annotated with domain predictions, such as the pfam annotations displayed here. *C. Predicted Alignment Error*: The predicted alignment error, measured in Ångstroms, is an AlphaFold metric. In Foldy it is rendered in the PAE tab with Plotly [16]. This displayed heatmap shows the predicted alignment error for the pik2 homodimer. The relatively low PAE between the KS-AT portions of the two peptides indicates that the two KS-ATs rigidly dimerize, while the comparably high PAE between the KRs and the rest of the structure indicate that the KRs move freely and independently. *D. Contact Probability*: The contact probability is derived from the internal AlphaFold distance probability distribution function, and can be interpreted as the likelihood that two residues are within 8Å of each other. For residues between disparate chains, this might suggest a contact point between two interacting domains or peptides. The 8Å contact probability is rendered in Foldy on the Contacts tab with Plotly [16]. Displayed is the contact probability for the predicted structure of the pik2 homodimer. Yellow dots in off-diagonal blocks suggest predicted interacting or contacting residues between the two peptides.

shown in Fig 2A, which shows the Foldy prediction for the structure of the homodimer of the polyketide synthase pik2. It also has NADPH as a cofactor, shown in a pose predicted by Auto-Dock Vina within the displayed bounding box around the AT domain. By default the tool panel displays the amino acid sequence, and Pfam [12] domain annotations can be overlaid on both the structure and sequence(s) (as shown in Fig 2B). A number of actions are available to users through the tabs in the tool panel. For example, users can predict residue interactions and complex formation using contact probability maps (Fig 2D) [6]. Users can segment proteins into domains and predict inter-domain flexibility using the Predicted Alignment Error (PAE, Fig 2C) [3]. Additionally, users can dock small molecule ligands with AutoDock Vina or DiffDock by specifying the SMILES string and optionally a bounding box around a residue [13,14].

More details about these views including screenshots are available at https://github.com/JBEI/foldy/blob/v1.0.1/docs/interface.md.

## Deployment options

Foldy is made up of services: a frontend, a backend, workers, and a few databases. The frontend service runs nginx and serves a compiled React website. The backend service runs Gunicorn and serves a Flask-REST interface. The worker service runs Flask RedisQueue workers, which imvoke all the bioinformatics tools via bash scripts. A Postgres database stores users, their structures, docked ligands, and the status of the invoked command line tools. A Redis

database stores a queue of work items, each corresponding to a command line invokation. Either object storage (e.g., Google Cloud Storage) or a disk store the raw outputs of each bioinformatics tool, depending on the deployment option. Finally, each bioinformatics tool's database is stored on a disk accessible to the worker services. The architecture is described at https://github.com/JBEI/foldy/blob/v1.0.1/docs/architecture.md.

These services can be configured in many different ways, and we provide instructions and helper scripts for two fully featured deployments, one which is quicker and the other more scalable. "Foldy-in-a-box" takes less than 10 minutes of work to set up, as determined by one non-computational user's experience. Setup requires reserving a large virtual machine on Google Cloud (eg, an Nvidia A100 GPU, 3TB disk, 50GB memory), and executing one command. The command downloads and runs a setup script that deploys all Foldy services with a Docker Compose file, configures that Compose file as a systemd service, and downloads the databases for the bioinformatics tools. The "Helm" deployment is fully horizontally scalable—Foldy services are run on Kubernetes, and each service is scaled up and down to match user demand. The Kubernetes services are managed by Helm, but some databases and various permissions need to be configured manually. There are also instructions for a local deployment for developers, which excludes the worker service, and is useful for development of the Foldy user interface. Instructions and further explanation for these three deployments are available at https://github.com/JBEI/foldy/tree/v1.0.1/deployment. Many other configurations are also possible.

## Scalability

By default, Foldy uses cloud compute for all tasks, meaning any lab or institution, regardless of their hardware, is able to set up Foldy. In the Foldy-in-a-box deployment, all compute tasks are run serially on the virtual machine. In the Helm deployment, jobs are run on ephemeral machines that are spawned when work tasks are queued and deleted when the work is complete. The work tasks are tracked in a queue (implemented with RedisQueue), and the worker machines are automatically created and destroyed by Kubernetes (implemented with Prometheus and KEDA). Importantly, each worker machine can be provisioned with high-memory GPUs, enabling the prediction of large protein structures.

Groups with access to their own compute resources, including compute clusters, can run additional worker threads on their own machines. To run a worker on a local compute resource which supports docker, one can run the "worker" docker image on the machines with the appropriate flags, and set up tunnels to the cloud databases. To use local compute resources which don't support docker, as is the case on some clusters, one can create bash scripts which execute each tool. For example, to run AlphaFold jobs on a university cluster which does not support docker, one must create a variant of "worker/run_alphafold.sh" which imvoke the local AlphaFold installation.

Additionally, groups with large personal machines (e.g. a 3TB disk, 50GB memory, and a GPU) can run Foldy locally, using a modified version of the "Foldy-in-a-box" docker compose file.

## Results

To underscore the wide utility of this tool, we touch on three projects facilitated by Foldy where Foldy was an easier-to-use solution than other open-source tools.

### Engineering the substrate preference of a fatty acyl-AMP ligase

One researcher was interested in changing the substrate preference of a long-chain fatty acyl-AMP ligase (FAAL) from long-chain to medium-chain, up to eight carbons long (C8-AMP).

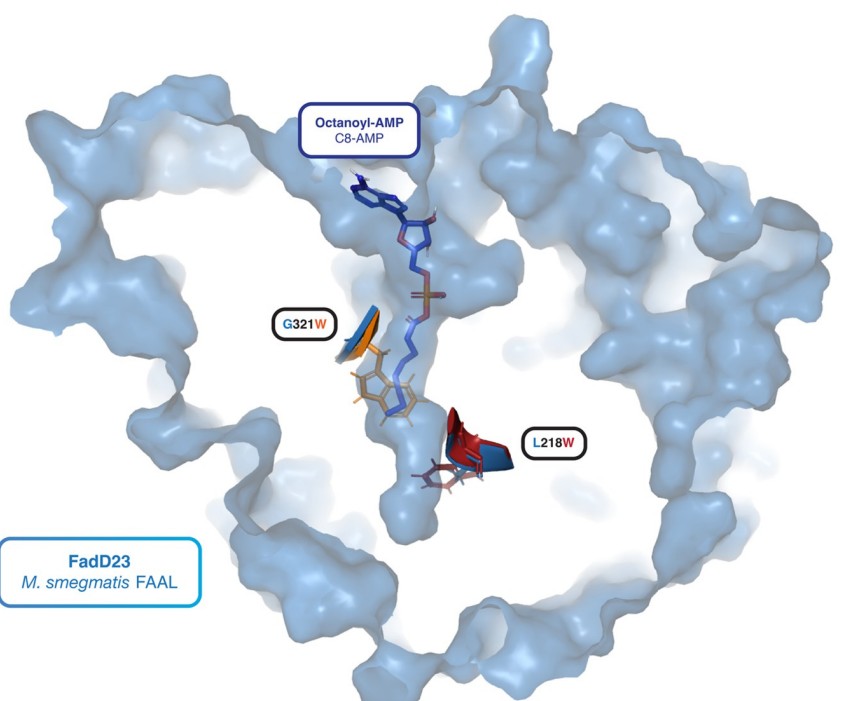

**Fig 3. A fatty acyl-AMP ligase with substrate docked.** The structure of the wild type FadD23 from *M. smegmatis* as predicted by AlphaFold in Foldy is shown in blue, rendered in pymol. Within the active site is the pose of octanoyl-AMP docked as predicted by AutoDock Vina in Foldy. This pose was chosen from the ensemble of predicted poses based on biochemical information about the active site. Two notable residues are highlighted: G321 and L218. The wild type backbone and side chain are displayed for both in blue. Displayed in orange and red are the same residues taken from two different structures which are mutants of the wild type. On top is residue 321 from the G321W mutant's structure as predicted by AlphaFold. This mutant seems to occlude the proper insertion of the C8 ligand. Below is residue 218 from the L218W mutant's structure as predicted by AlphaFold. This mutation seems to allow insertion of the C8 ligand but would occlude insertion of longer ligands like the C10-AMP. Based on these three structures, L218W is a good candidate for making an FAAL which prefers shorter chain fatty acyl AMPs while preserving activity for the C8 chains.

They used Foldy to predict the structure of wild type protein, FadD23 from *M. smegmatis*'s structure, and used AutoDock Vina to dock octanoyl-AMP in the active site tunnel. They used the structure to suggest two point mutations—G321W and L218W—which they hypothesized would obstruct the substrate tunnel and shorten chain-length preference of the FAAL, while still allowing docking of the C8 ligand. They then predicted the structures of those mutants with Foldy. They found that the active site of the G321W mutant was too short to accommodate C8 ligands, and found that the L218W mutation would occlude C10 or larger while allowing C8 (Fig 3). The structures and their ligands are available at https://foldy.lbl.gov/tag/msmeg_faal.

Neither AlphaFold-Colab nor ColabFold support the quick iteration of design & test of mutant proteins by running both AlphaFold predictions and small molecule ligands in the same tool.

## Prediction of chimeric megasynthase function

Foldy was used to evaluate AlphaFold Multimer's ability to predict chimeric polyketide synthase production. Nava et al [17] evaluated whether a chimeric PKS's predicted structure is indicative of its production titer. A total of 144 interacting KS-AT / ACP pairs from a seminal

paper about module swaps [18] were co-folded as trimers (KS-AT/KS-AT/ACP) with Alpha-Fold Multimer in Foldy, and the protein contact probability map was used as a proxy for likelihood of protein interaction, as described by Humphreys et al [6]. The authors did not find a correlation between predicted ACP/KS-AT interaction and titer, but found the structures informative and worth more investigation. The structures of the PKS dimers are available at https://foldy.lbl.gov/tag/menzelladimer.

Neither AlphaFold-Colab nor ColabFold support the prediction of large megasynthase complexes (1000+ amino acid) without a Google Colab paid membership or dedicated virtual machine or hardware, and do not make it easy to store and download large numbers of structures.

## Characterizing domains of unknown function

Metabolic engineers used Foldy to augment their other data to predict the function of dozens of domains of unknown function (DUFs) in *P. putida*, including DUF1302 and DUF1329. The researchers were studying the flexible carbon metabolism of *P. putida* using data from multiple random barcode transposon-site sequencing (RB-TnSeq) experiments. Thousands of the genes that had significant fitness phenotype in some condition had domains of unknown function (DUFs), including PP_0765 and PP_0766. These two proteins were essential for the metabolism of Tween20, but their protein families, with pfam IDs DUF1302 and DUF1329 respectively, had unknown function. Prior RB-TnSeq experiments show the function of PP_0765 and PP_0766 correlates with two other genes which are essential for growth on Tween20: a periplasmic protein (PP_2018) and a multidrug efflux transporter (PP_2019) [19].

In this case study, the metabolic engineers used AlphaFold to predict the likelihood of protein-protein interactions by co-folding multiple proteins, and using AlphaFold's residue-residue distance prediction as a proxy for probability of two residues interacting, as was demonstrated by Humphreys et al [6]. Foldy facilitates this type of analysis by calculating the maximum contact probability between different peptides in the prediction (Fig 4A). This information is visible in the Contacts tab of the Structure View. PP_0766 is predicted to interact with both PP_2019 and PP_0765, which, along with PP_0765's distinctive transmembrane beta barrel structure, suggest that PP_0766 may be localized to the periplasm, and PP_0765 may be localized to the outer membrane (Fig 4). Altogether this suggests that PP_0765 and PP_0766, which were previously suspected of being involved in hydrolase activity [19], may actually be components of a novel transport system. The predicted structures of some of the *P. putida* DUFs and the complex structure predictions are publicly visible at https://foldy.lbl.gov/tag/putida.

Neither AlphaFold-Colab nor ColabFold support the prediction of large complexes (1000 + amino acid) without a Google Colab paid membership or dedicated hardware.

## Discussion

This tool greatly facilitates research because it makes complex tools accessible. There has been much work done to make advanced structural biology tools more accessible, both open- and closed-source, and we believe Foldy is the most featureful open-source tool which can run AlphaFold. The LBL Foldy instance (https://foldy.lbl.gov) has been used by 55 researchers across 6 labs, to predict 6493 structures and dock 2754 ligands.

## Comparison to other tools

We compare Foldy to other current tools for doing modern structure prediction (Table 1). The source code for open source software is available to view, clone, and use with little restriction.

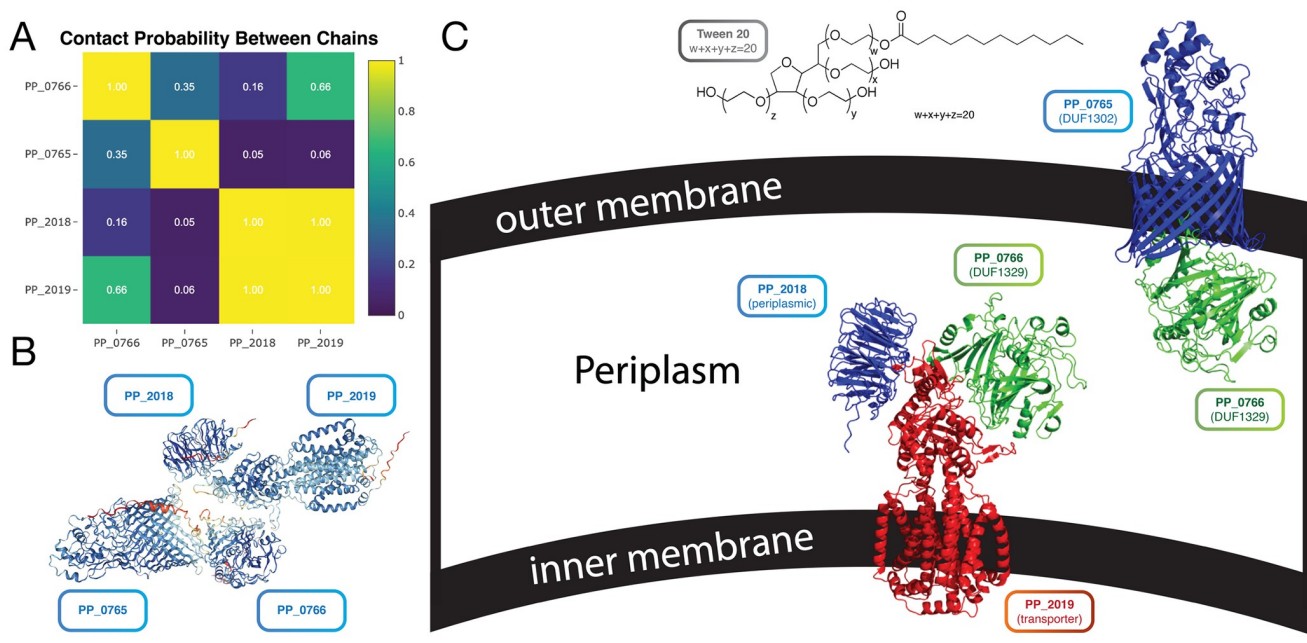

**Fig 4. Putative complex formation of two domains of unknown function in *P. putida*.** Four genes known to be essential for metabolism of Tween-20 were co-folded with AlphaFold in Foldy. *A. Peptide-Peptide Contact Probability*: As done in Humphreys et al [6] Foldy approximates the peptide-peptide contact probability as the maximum contact probability of any residues between two cofolded peptides. The matrix of peptide-peptide contact probabilities is shown in the Contact tab of the Structure View page. This contact probability matrix shows that maybe: PP_0766 and PP_2019 weakly interact, PP_0766 and PP_0765 weakly interact, and PP_2019 and PP_2018 strongly interact. *B. Predicted Interactions of Phenotypically Related Proteins*: The four proteins whose knockouts have related phenotypes were folded in Foldy, and although all four cluster closely, the peptide-peptide contact probability map indicates that not all four interact directly with each other. By using biochemical knowledge about these proteins, we are led to suspect that there are two different complexes which can form. *C. Novel Hypothesized Substrate Transport System*: Two DUFs in *P. putida*, previously hypothesized to have hydrolase activity may actually be involved in substrate transport: PP_0765 (DUF1302, top blue) and PP_0766 (DUF1329, top green, bottom green). PP_0765 has the characteristic beta-barrel of a membrane protein, and shows high likelihood of forming a complex with PP_0766 (top). Additionally, PP_0766 is predicted to form a heterotrimer with PP_2018 (bottom blue) and PP_2019 (bottom red).

Cost is difficult to compare for a few reasons: cloud compute resources are more expensive than local hardware, but easier to acquire; closed source software is often free for a trial and subject to variable pricing in the future. But generally, open source software is cheaper to run because users only pay for compute resources with no premium for software developers and corporate profits. For example, predicting the structure of a large (~4000AA) structure on

**Table 1. Comparison of different structure prediction tools.**

| Tool | Open Source | Terminal-free access | Stores & manages structures | Docking & Annotations | Size Upper Bound |
|---|---|---|---|---|---|
| AlphaFold | ✓ | | | | 6000AA |
| ColabFold (free version) | ✓ | ✓ | | | 1000AA |
| ColabFold (Pro or reserved A100) | ✓ | ✓ | | | 6000AA |
| BioNeMo | | ✓ | ✓ | ✓ | *Unavailable* |
| LatchBio | | ✓ | ✓ | ✓ | 5000AA |
| NeuroSnap | | ✓ | ✓ | ✓ | 5000AA |
| Benchling | | ✓ | ✓ | | 1500AA [9] |
| Foldy on Helm | ✓ | ✓ | ✓ | ✓ | 6000AA |
| Foldy-in-a-Box with a T4 GPU | ✓ | ✓ | ✓ | ✓ | 1000AA |
| Foldy-in-a-Box with an A100 GPU | ✓ | ✓ | ✓ | ✓ | 6000AA |

Foldy deployed with Helm costs $5-$10, as determined by Google Cloud Billing console. Whereas Latch Bio currently gives some number of free structure predictions to academic labs, and then charges ~150 credits per large structure task, which is equivalent to ~$150.

Some tools offer more services than just structure prediction, including storing and managing structures, and running downstream analyses like domain annotations and small molecule docking. Foldy is the only open-source tool which can store and manage AlphaFold predicted structures, and the only open source tool which can both dock molecules and run AlphaFold for structure prediction.

Finally, different AlphaFold hardware setups allow different maximum structure sizes. We determined a rough upper bound by running a series of structure predictions from 1000–7000 amino acids, and we report the smallest structure which failed to fold. For example, Foldy on Helm was able to predict the structure for an input sequence with 5000 amino acids, but failed on structures of size 6000 and above, so we report a size upper bound of 6000 amino acids. Note that we used the reported upper bound for Benchling. The Foldy test structures are publicly available at https://foldy.lbl.gov/tag/testmaxsize.

## Conclusion

Foldy is easier to use than other open-source AlphaFold implementations and addresses some of their limitations [1,7]. User experience studies indicate that small improvements in the enjoyability of a tool may have significant effects on tool use [20]. Foldy will increase biologists' productivity by increasing the adoption of computational structure tools. The adoption of Foldy, including by those without large compute clusters or GPUs, will make high-accuracy protein structure prediction more accessible.

## Author Contributions

**Conceptualization:** Jacob B. Roberts, Alberto A. Nava, Matthew R. Incha.

**Data curation:** Jacob B. Roberts, Alberto A. Nava, Allison N. Pearson, Matthew R. Incha.

**Formal analysis:** Alberto A. Nava, Matthew R. Incha.

**Funding acquisition:** Jay D. Keasling.

**Investigation:** Jacob B. Roberts, Allison N. Pearson, Matthew R. Incha.

**Methodology:** Jacob B. Roberts, Matthew R. Incha.

**Project administration:** Jay D. Keasling.

**Resources:** Jay D. Keasling.

**Software:** Jacob B. Roberts, Alberto A. Nava, Matthew R. Incha, Melody Ma, Abhay Rao.

**Supervision:** Jay D. Keasling.

**Validation:** Jacob B. Roberts, Alberto A. Nava, Allison N. Pearson, Matthew R. Incha, Luis E. Valencia.

**Visualization:** Jacob B. Roberts, Alberto A. Nava, Allison N. Pearson, Matthew R. Incha.

**Writing – original draft:** Jacob B. Roberts, Alberto A. Nava, Allison N. Pearson, Matthew R. Incha, Jay D. Keasling.

**Writing – review & editing:** Jacob B. Roberts, Alberto A. Nava, Allison N. Pearson, Matthew R. Incha, Melody Ma, Jay D. Keasling.

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
