## [Decision Letter · Decision Letter 0]

21 Jul 2023

Dear Dr. Keasling,

Thank you very much for submitting your manuscript "Foldy: an open-source, cloud-based web application for interactive protein structure analysis" for consideration at PLOS Computational Biology.

As with all papers reviewed by the journal, your manuscript was reviewed by members of the editorial board and by several independent reviewers. In light of the reviews (below this email), we would like to invite the resubmission of a significantly-revised version that takes into account the reviewers' comments.

We cannot make any decision about publication until we have seen the revised manuscript and your response to the reviewers' comments. Your revised manuscript is also likely to be sent to reviewers for further evaluation.

Sincerely,

Christos A. Ouzounis

Academic Editor

PLOS Computational Biology

Arne Elofsson

Section Editor

PLOS Computational Biology

Reviewer's Responses to Questions

**Comments to the Authors:**

Reviewer #1: This is potentially a very useful application to allow users to predict AlphaFold structures in their local servers or via Google clouds. I created issues in Foldy GitHub pages when I had problems. However, after three weeks, I still can not use it to predict AlphaFold structures. I am sure most users will have the same experience if they try it.

The authors should spend significant effort to make sure users can use the methods to run their own structure prediction. I have the following comments.

1. If the local deployment is used for testing purpose and can not predict structures, authors should either describe how to use it for testing, or don't mention it all. My first impression is that I can use it to predict structures, but can't use the analysis tools.

2. The steps to set up production deployment could be mentioned in the main paper. They are more important than the use cases. The supplementary should describe in more details about the steps so that other people can reproduce it. It should include some alternatives if something goes wrong.

3. I spend three weeks and only get to step 6. Here are some comments about the application:

1) Typo "]" in the file foldy/values.yaml: postgresql://postgres:${GOOGLE_SQL_DB_PASSWORD}@]${GOOGLE_SQL_DB_PRIVATE_IP}/postgres

2) Add the step to install yq. I used “pip3 install yq” and it actually is “jq”. I have to replace “yq eval” with “yq -r”.

3) I used “us-east1” from the beginning. So “us-central1”/”us-central1-c” should be a variable.

4. The value of the paper is to allow users to use your method to set up their own servers. I am sure they will have many questions. A timely response is crucial.

5. If possible, you could include ESMFold in local deployment.

Reviewer #2: The presented paper offers an accessible web interface for protein structure prediction and ligand docking, employing advanced AI-based tools in structural biology, including AlphaFold and DiffDock. The tool can either be hosted on the provided cloud server or local devices, expanding its accessibility to biologists, even those with limited software expertise. The showcased case studies underscore the wide utility of this interface for ligand binding prediction, predicted structure evaluation, and protein-protein interaction prediction. This paper introduces a powerful tool for protein structure analysis.

Considering the potential impact of this article, the manuscript necessitates significant revisions as follows:

Major Revisions:

1. Results – Unsupported Advantage Claim: The assertion in the abstract that "Foldy enables scientists to predict the structure of proteins and complexes up to 3000 amino acids" lacks supporting evidence in the results, calling into question the stated benefit of the tool.

2. Results – Absence of Benchmark Comparison: The paper acknowledges ColabFold and AlphaFold-Colab as benchmark applications for protein structure prediction, and AutoDock Vina and DiffDock for ligand docking, yet fails to provide a rigorous quantitative comparison to these benchmarks. It is crucial for the Scalability section to convincingly demonstrate how Foldy overcomes these benchmarks' limitations.

3. Discussion – Informal Case Studies: The case studies lack clear objectives, resembling experimental notes rather than comprehensive research investigations. Key details like target protein names, sequence lengths, processing methods, and benchmark comparison results should be explicitly presented. Furthermore, it appears that the case studies could potentially be replicated using benchmark tools, undermining the claim of Foldy's unique capabilities.

Minor Revisions:

1. Results – Figure 1: Detailed information about the dashboard in relation to the third case study would enhance understanding. A clear reference to this figure within the case study should also be considered.

2. Results - Figure 2: NADPH is mentioned only in this figure and lacks corresponding references in the main text.

3. Results - Figure 3 - Detailed Process Flow: Depicting the process flow for specific projects like protein-protein interaction or protein-ligand interaction in Figure 3, and outlining differences when executing various projects would be beneficial. Also, clarifying the values in the figure, such as the color bar ranges, and their meanings would enhance understanding. A dedicated method section can be used to illustrate the processing pipeline.

4. Results - Placement of Case Studies: It is recommended to relocate the case studies to the results section for a more logical flow of content.

5. Discussion - Figure 4: It's unclear how the visualization was achieved. Is this the view from within Foldy?

6. Discussion - Figure 5: Details on how membrane proteins were modelled and protein-protein interactions predicted when membrane protein is involved should be provided, especially as PP_0765 and PP_2019 are shown in Figure 5.

Reviewer #3: This manuscript described a web server that assemble multiple state-of-the-art AI-based tools in structural biology, including AlphaFold and DiffDock. There are also existing web servers providing such services for proteins. For example, Neurosnap (https://neurosnap.ai/) provides web services including AlphaFold, DiffDock, ESMFold, LightDock, and etc,. There are even more web servers for protein-ligand binding. The manuscript did not demonstrate the advantages of their own server comparing to the others thoroughly. Besides, the provided application requires a sophisticated local installation to run the web tools instead of providing a directly usable web server, it increased the barrier to access the services to life science researchers. The platforms might be user-friendly, but the way to set up the platforms is not so friendly.

**Have the authors made all data and (if applicable) computational code underlying the findings in their manuscript fully available?**

Reviewer #1: Yes

Reviewer #2: Yes

Reviewer #3: Yes

PLOS authors have the option to publish the peer review history of their article (what does this mean?). If published, this will include your full peer review and any attached files.

Reviewer #1: **Yes: **Jiyao Wang

Reviewer #2: **Yes: **Jiangguo Zhang

Reviewer #3: No
---

## [Decision Letter · Decision Letter 1]

2 Dec 2023

Dear Dr. Keasling,

Thank you very much for submitting your manuscript "Foldy: an open-source web application for interactive protein structure analysis" for consideration at PLOS Computational Biology. As with all papers reviewed by the journal, your manuscript was reviewed by members of the editorial board and by several independent reviewers. The reviewers appreciated the attention to an important topic. Based on the reviews, we are likely to accept this manuscript for publication, providing that you modify the manuscript according to the review recommendations.

Just fix the suggested minor changes and ensure that everything works.

Sincerely,

Arne Elofsson

Section Editor

PLOS Computational Biology

Arne Elofsson

Section Editor

PLOS Computational Biology

Just fix the suggested minor changes and ensure that everything works.

Reviewer's Responses to Questions

**Comments to the Authors:**

Reviewer #1: Thank you for taking your time to add the easier option Foldy-in-a-box deployment. It's also much clearer that the development option is for interface only.

Reviewer #2: Foldy is a useful interface that bridges the gap between experimental structural biologists and advanced AI-based protein structure prediction and docking models. Its user-friendly design and minimal computational resource requirements make it especially beneficial for structural biologists who may have limited coding experience or computational resources.

The revised version is much better prepared for publication. Each example is better introduced, the hypothesis is clear, and the method is easy to understand and reproduce. The inclusion of the comparison table shows the advantage of Foldy clearly. The updated instructions on GitHub are better described and easier to follow. However, there are minor issues that need to be addressed before it can be accepted for publication.

Issue:

When I tried to deploy Foldy-in-a-box on Google Cloud, the specified configuration, a n1-highmem-8 VM instance with 1 nvidia-tesla-t4 accelerator(s) was unavailable. It would be nice if there were alternative configurations for users to choose from.

Minor Revisions:

Figure 3: The caption mentions "L281W" while the figure shows "L321W."

Figure 4c: The left "PP_0765" should be "PP_0766," and the bottom "PP_0766" should be "PP_2019."

Table 1: The "???" on the row of BioNeMo is unclear. Does it mean that it is unavailable to analyze?

**Have the authors made all data and (if applicable) computational code underlying the findings in their manuscript fully available?**

Reviewer #1: Yes

Reviewer #2: Yes

PLOS authors have the option to publish the peer review history of their article (what does this mean?). If published, this will include your full peer review and any attached files.

Reviewer #1: **Yes: **Jiyao Wang

Reviewer #2: **Yes: **Jiangguo Zhang

Figure Files:

Data Requirements:

Reproducibility:

References:

---

## [Decision Letter · Decision Letter 2]

5 Jan 2024

Dear Dr. Keasling,

We are pleased to inform you that your manuscript 'Foldy: an open-source web application for interactive protein structure analysis' has been provisionally accepted for publication in PLOS Computational Biology.

Best regards,

Christos A. Ouzounis

Academic Editor

PLOS Computational Biology

Arne Elofsson

Section Editor

PLOS Computational Biology

Reviewer's Responses to Questions

**Comments to the Authors:**

Reviewer #2: I am pleased to recognize that the manuscript 'Foldy: an open-source web application for interactive protein structure analysis' has effectively addressed all previously raised concerns. The revisions enhance the manuscript, illustrating Foldy's significant role in helping structural biologists to utilize advanced AI models with limited coding expertise and computational resources. This work substantially contributes to the field of structural biology. Accordingly, I recommend this manuscript for publication.

**Have the authors made all data and (if applicable) computational code underlying the findings in their manuscript fully available?**

Reviewer #2: Yes

PLOS authors have the option to publish the peer review history of their article (what does this mean?). If published, this will include your full peer review and any attached files.

Reviewer #2: **Yes: **Jiangguo Zhang

---

## [Editor Report · Acceptance letter]

30 Jan 2024

PCOMPBIOL-D-23-00749R2 

Foldy: an open-source web application for interactive protein structure analysis

Dear Dr Keasling,

I am pleased to inform you that your manuscript has been formally accepted for publication in PLOS Computational Biology. Your manuscript is now with our production department and you will be notified of the publication date in due course.

With kind regards,

Lilla Horvath
